# Conformal Linguistic Calibration:
# Trading-off between Factuality and Specificity

**Zhengping Jiang**
Johns Hopkins University
`zjiang31@jhu.edu`

**Anqi Liu**
Johns Hopkins University
`aliu.cs@jhu.edu`

**Benjamin Van Durme**
Johns Hopkins University
`vandurme@jhu.edu`

## Abstract

Language model outputs are not always reliable, thus prompting research into how to adapt model responses based on uncertainty. Common approaches include: *abstention*, where models refrain from generating responses when uncertain; and *linguistic calibration*, where models hedge their statements using uncertainty quantifiers. However, abstention can withhold valuable information, while linguistically calibrated responses are often challenging to leverage in downstream tasks. We propose a unified view, Conformal Linguistic Calibration (CLC), which reinterprets linguistic calibration as *answer set prediction*. First we present a framework connecting abstention and linguistic calibration through the lens of linguistic pragmatics. We then describe an implementation of CLC that allows for controlling the level of imprecision in model responses. Results demonstrate our method produces calibrated outputs with conformal guarantees on factual accuracy. Further, our approach enables fine-tuning models to perform uncertainty-aware adaptive claim rewriting, offering a controllable balance between factuality and specificity.[1]

## 1   Introduction

Large language models (LLMs) can provide knowledge responses based on their comprehensive training sets [48, 59, 78], but these responses may not be trustworthy [41, 26], and can be stated in an over-confident tone [42]. One solution is to communicate a calibrated uncertainty measure to the user, so as to enable more informed decision-making [80]. This uncertainty information can be expressed in different ways, which we hereby classify into two groups. The first set of approaches marks a response explicitly with the model's level of confidence [66, 12, 65]. When such uncertainty expressions are conveyed through text generation and are adjusted to faithfully represent accuracy of the response, this is called *linguistic calibration* [42, 3, 6, 69]. For example, a model uncertain in its response may state: "*Possibly* the largest city in the US is Los Angeles" (Fig 1).[2] However, how to interpret linguistic uncertainty can be unclear and subjective. Also, the outer structure introduced by linguistic calibration hampers the downstream task / evaluation [34], as it is unclear what special treatment should be given to these uncertain assertions. Another set of approaches have the model generate prediction sets instead of a single response, such that with high probability some are acceptable [50, 32, 55]. A special case of this is when a model abstains from responding when it is not confident [58, 28, 39, 45, 9, 18]. E.g., "I don't know" is trivially equivalent to predicting the

---

[1]Code available at: `https://github.com/zipJiang/CLC`.
[2]Answer by population is *New York City*, by area is *Sitka*.

39th Conference on Neural Information Processing Systems (NeurIPS 2025).

set of all possible answers. However, this breaks the normal human-model interaction paradigm, and the set prediction is not directly actionable [11].

To address these limitations, we propose a novel unifying view, *Conformal Linguistic Calibration*: having the model express uncertainty levels by generating less specific claims until they are confident. We achieve this by reinterpreting uncertainty quantifiers as set prediction using recent advances [19] in possible world semantics [30, 29]. At a high level, being more confident in a claim means that a subject would only regard it as false in counterfactual worlds that are drastically different from the actual one. These gradually less similar counterfactual worlds form a series of nested sets corresponding to the subject's belief [36, 63]. From this perspective, linguistic calibration can be seen as a form of *soft abstention*, whereas traditional abstention [28, 44] corresponds to committing to the universal set of all possible worlds. As shown in Figure 1, this allows us to improve the flexibility and controllability over an abstention-based approach. Also, our formulation naturally supports conformal prediction techniques [67, 1], offering probabilistic guarantees on factuality while producing confident, clear claims that are easy for downstream use. In summary, we:

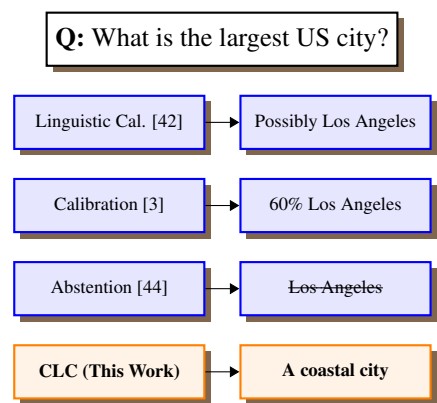

Figure 1: Distinct from previous approaches CLC allows generating a less precise but more confident statement.

1. Propose a new paradigm for model to communicate their uncertainty.

2. Provide an algorithmic instantiation of the process by probing semantic uncertainty [31], building nested subsets, and summarizing beliefs, which can be post-hoc calibrated through learn-then-test (LTT) [1].

3. Show that our approach enables a smooth trade-off between factuality and specificity [74]—for example, allowing a 7B model to outperform GPT-4o in factuality on a challenging QA dataset by answering less specifically [74].

4. Train an amortized, uncertainty-aware claim rewriter that explicitly trades off specificity for factuality following user instructions. We release both our rewriting model and dataset.

## 2 Related Work

**LLM Factuality Evaluation**   Large language models enable rich long-form generation [5, 46], yet hallucinations remain persistent [41, 26, 24]. To monitor factual precision, evaluation pipelines typically decompose model outputs into atomic claims and verify each against trusted sources or retrieval-augmented evidence [43, 71, 72]. Subsequent work sharpens these pipelines by improving claim extraction and coverage [73, 21, 27], but the focus remains on enumerating verifiably true statements. As a result, they offer limited guidance on how to modulate hedging or other forms of controlled imprecision that influence downstream usability [34].

**Conformal Prediction and Risk Control**   Risk-sensitive frameworks complement precision-centric evaluation by directly bounding the probability of erroneous outputs. Conformal prediction has been adapted to over-generation, abstention, delegation, and clarification settings in language applications [50, 28, 77, 20, 49, 17, 56], providing explicit coverage guarantees when the model defers, expands, or filters its answers. The closest approach to ours, Mohri and Hashimoto [44], drops uncertain claims to improve faithfulness, and Cherian et al. [10] refine the procedure via adaptive conformal prediction. We maintain the original answer space but rewrite claims so that conformal guarantees translate into interpretable hedging, allowing users to trade specificity for reliability while retaining coherent narratives.

# 3 Preliminaries

We focus on the setting where given a prompt $x$, a language model $\mathcal{L}$ generates response $y = \mathcal{L}(x)$ where $y \in \mathcal{Y}$. The overall goal is to find a post processor $\mathcal{T} : \mathcal{Y} \to \mathcal{Y}$, that ensures a probabilistic guarantee for a user specified probability $\alpha \in (0, 1)$:

$$\mathbb{P}(\mathcal{T}(y) \text{ is factuality correct}) \geq 1 - \alpha.$$

One particular challenge is to find a $\mathcal{T}$ that will almost always work without too many constraints on $\mathcal{Y}$. For example Mohri and Hashimoto [44] shows that for longer generations one can come up with a simple solution for $\mathcal{T}$ where one can drop a subset of claims, a process they called *back-off*, but this will not work for more atomic generations like in question answering, as *back-off* is highly restrained by explicitly stated claims. Yet atomic-claim-level operation is very desirable, as previous works have demonstrated the benefit of decomposition for various fact verification problems [43, 27, 53, 73, 64]. To allow being less specific beyond the surface form, we need a more sophisticated process for identifying plausible alternatives to the input claim to guide post-processing. We now describe how the post-processing step $\mathcal{T}$ can be formalized in terms of *belief*. This helps connect back to linguistic calibration, and we provide a way to achieve our desired guarantee despite the additional complexity our proposal introduces.

## 3.1 Possible World Semantics

The notion of a *possible world* has a long tradition in philosophy, described as the "limit of a series of increasingly more inclusive situations" [16]. Following Kripke [29], modal expressions are evaluated by asking which alternative situations remain accessible from the world we currently inhabit. We therefore treat a claim as the set of worlds in which it remains true: relaxing the claim (i.e., hedging) enlarges that set, lowering the risk of factual error while preserving informativeness.

Formally, a Kripke model $M = (W, R, \Vdash)$ consists of a set of possible worlds $W$, an accessibility relation $R$ where $R(w, u)$ indicates that world $u$ is accessible from $w$, and a satisfaction relation $\Vdash$ that evaluates formulas at a world. These ingredients let us define the familiar necessity and possibility operators for a claim $c$[3].

**Definition 3.1** (Necessity). Under $M$ at $w \in W$, a necessitation $\Box c$ is true iff $\forall u \in W$ such that $R(w, u)$, we have $u \Vdash c$.

The "possibly" operator $\Diamond$ then follows in the usual way:

**Definition 3.2** (Possibility).

$$\Diamond c := \neg \Box \neg c.$$

Accessibility relations organize the worlds around $w$ into concentric *spheres* that reflect increasingly remote departures from what the agent regards as normal or likely. A claim is necessary within a sphere if it holds throughout that sphere, and merely possible if it holds in at least one world contained in it. Different modals can therefore adopt different relations: Hintikka [23] extends Kripke semantics to belief with a plausibility-based relation $R_B$ calibrated to empirical constraints on human belief. To model graded confidence, Goodman [19] combines such relations with similarity orderings, where $u \succeq_{A,w} v$ means that agent $A$ judges $u$ to be at least as similar to $w$ as $v$ is. The resulting nested family

$$S_{A,w} := \Big\{ \{u \in W \text{ s.t. } u \succeq_{A,w} v\} \text{ s.t. } v \in W \Big\}$$

collects precisely these spheres [35]. Given a subjective probability $\mathrm{Pr}_{A,w}$, one can associate each confidence level $d$ with the set of worlds

$$\mathrm{Conf}_A^d(w) = \bigcap \Big\{ p \in S_{A,w} \text{ s.t. } \mathrm{Pr}_{A,w}(p) > d \Big\},$$

which is itself nested and can be repackaged as an accessibility relation that yields graded confidence modals. Viewing claims as nested sets of worlds thus reveals a direct bridge between claim generation and set prediction, allowing us to adapt conformal prediction techniques for risk control in Section 4. We defer richer illustrations of this perspective to the expanded revision, where additional space is available.

---

[3]Throughout this work we view a claim as a *nullary predicate* that does not take arguments.

## 3.2 Conformal Prediction

**Split Conformal Prediction**    Vovk et al. [67], Shafer and Vovk [61], Papadopoulos [47] provide standardized tools to construct prediction sets that provide coverage guarantees. Concretely, using a calibration dataset $(X_i, Y_i)_{i=1,...,n}$, split conformal prediction gurantees that for i.i.d. sample $(X_{\text{test}}, Y_{\text{test}})$ with a prediction set $C(X_{\text{test}}) \in 2^{\mathcal{Y}}$, then for any designated target threshold $\alpha \in (0, 1)$

$$\mathbb{P}(Y_{\text{test}} \in C(X_{\text{test}})) \geq 1 - \alpha. \tag{1}$$

Following the view of Gupta et al. [22], the split conformal prediction procedure starts from a sequence of nested candidate output sets, and use calibration data to select the correct level of in the nested set until the coverage guarantee is achieved.

However, this approach requires that the prediction sets to select from are nested, or similarly, the non-conformity score, or equivalently the set construction is by thresholding on a sequence of monotonous non-conformity scores [2]. Instead of using the quantiles of a scoring function, a more general extension of conformal prediction called Learn-Then-Test (LTT) [1] relies on hypothesis testing to identify a viable region to control any hyper-parameter sensitive risk.

**Learn-Then-Test (LTT)**    extends conformal prediction to find a hyperparameter configuration $\lambda$ control the expectation of any risk function $R$ such that

$$\mathbb{P}(\sup_{\lambda \in \Lambda} R(\lambda) \leq \epsilon) \geq 1 - \alpha.$$

Unlike conformal risk control [2], LTT does not rely on the risk function being monotonous on $\lambda$. To achieve this, LTT associates the null hypothesis: $\mathcal{H}_\lambda : R(\lambda) > \epsilon$ to each configuration $\lambda$, and calculates a conservative p-value [4] for each of the hypotheses, from which the LTT guarantee directly follows.

## 4 Methods

In this section, we outline the procedure for deriving the risk-controlled process $\mathcal{T}$, as illustrated in Figure 2. Building on our previous discussion in Section 3.1, the objective is to leverage conformal prediction techniques to manage the risk associated with adherence to each level of the nested sphere. The underlying intuition is that the confidence level $V$ can equivalently be represented by a claim $\tilde{c}$ that *describes* the sphere $V^4$. We formalize this property as follows:

**Definition 4.1** (Description). A claim $\tilde{c}$ is said to describe a sphere $V$ iff

$$(\Box_V \tilde{c}) \wedge (\neg \Diamond_{W \setminus V} \tilde{c}). \tag{2}$$

Thus given a set of source claims $\{c_1, c_2, \ldots, c_N\}$, we aim to rewrite them to a set of less specific (or imprecise) claims $\{b_1, b_2, \ldots, b_N\}$ that each properly describes its corresponding possible world set (sphere) $\{W_1, W_2, \ldots, W_N\}$ such that for any designated target threshold $\alpha \in (0, 1)$

$$\mathbb{P}\left[\Diamond_W c | W = W_c\right] \geq 1 - \alpha,$$

which matches the guarantee in Equation (1). However, there is no practical way to directly evaluate $\Box_b c$ as there is no way to constructively derive the set $W_b$. In this section, we describe a data processing pipeline that for each claim $c$ derives a sequence of candidate target claims $b_c^1, b_c^2, \ldots, b_c^K$ such that the corresponding possible world sets in theory satisfy the nested assumption $W_c^1 \subset W_c^2 \subset \cdots \subset W_c^K$.

### 4.1 Less Specific Rewriting

In this section, we outline the pipeline to derive a less specific claim from a source answer string, with the goal of having the factuality risk of the generated claim properly controlled. We call this process claim rewriting. Our claim data is sourced from QA datasets, motivated by recent advancements in decomposition [72, 21], which demonstrate that complex text can be broken down into relatively simple, targeted questions [8, 76]. Furthermore, prior work has established that conversion between questions and claims is feasible and natural [7].

---

[4]With a slight abuse of notation, we denote the necessity operator associated with modal logic $M = (W, R_U, \Vdash)$ by $\Box_U$, where the accessibility relation is defined as $R_U(w, u) := \mathbb{I}[u \in U]$. A similar convention applies to the possibility operator $\Diamond$.

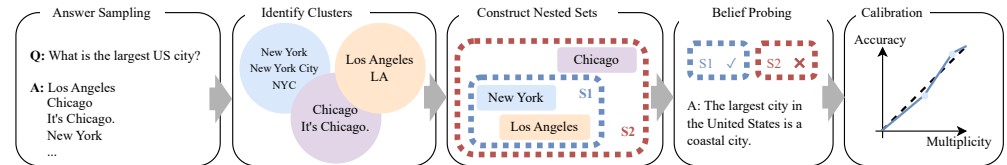

Figure 2: The overview of our conformal linguistic calibration approach. Instead of relying on direct operations like subclaim drop-off, we probe the model's internal belief by semantically clustering all sampled answers into nested sets, and writing less specific claims that are associated to each answer set level through Equation (2).

**Answer Sampling**  Given a question $q$, we repeatedly sample $K$ responses from a language model $L$, denoted as $a_k^q$.[5] This approach is a standard technique for estimating predictive uncertainty in natural language generation [70, 31, 3].

**Identifying Clusters**  From the sampled answer set $\{a\}$, we identify semantically unique clusters $z_{i=1}^Q$ [31]. Previous approaches typically employ an entailment model or its extensions to establish an equivalence relationship—such as bidirectional entailment—to induce clusters [31, 38]. However, while the number of clusters identified using this method serves as a useful indicator of response uncertainty, the clustering itself tends to be quite noisy. This is partially due to the inherent difficulty of fully defining an equivalence relationship in real-world scenarios. When operationalized through a Natural Language Inference (NLI) model [13, 40], such relationships often suffer from a lack of deep semantic understanding and excessive sensitivity to surface variations.

To address these issues, we instead use a single LLM call to directly generate a list of identifiable unique answer cluster names from $\{a\}$. The details of this prompt, along with other relevant prompts, are provided in Appendix A.

**Estimating Answer Multiplicity**  Given the answer set $\{a\}$ and a set of representative unique answers $Z \subset \{a\}$, we estimate the multiplicity of each cluster representative $z_i \in Z$ by counting the number of answers that are semantically equivalent to the corresponding cluster name. The cluster assignment $\delta_i(a_k)$ is determined using a similarity metric $s(\cdot, \cdot) \in [0, 1]$, which assigns $a_k$ to the $i$-th cluster such that

$$\delta_i(a_k) = \begin{cases} 1 & \text{if } i = \arg\max_{\tilde{i} \in |Q|} s(a_k, z_{\tilde{i}}) \\ 0 & \text{otherwise} \end{cases} \quad .$$

with arbitrary tiebreak. And the multiplicity is simply given through $m(z_i) = \sum_{k=1}^K \delta_i(a_k)$.

We acknowledge that while accurately estimating $m(c)$ is important, minor noise in answer cluster assignment does not compromise the validity of the pipeline. This is because the entire rewriting process is calibrated using the Learn-Then-Test (LTT) framework, as described in Section 4.2.

**Constructing Nested Cluster Sets**  In line with the approach of Wang et al. [70], we employ majority voting to identify the most confident answer, which we designate as the original answer $\tilde{z}_1$ where

$$\tilde{z}_1 = \arg\max_{i \in |Q|} m(z_i),$$

$$\tilde{Z}_1 = \{\tilde{z}_1\}.$$

We then incrementally add other clusters into the set to form a set of nested cluster sets

$$\tilde{Z}_1 \subset \tilde{Z}_2 \subset \cdots \subset \tilde{Z}_N.$$

To better align with the interpretation described in Section 3.1, we prioritize clusters that are semantically closer to the representative of the most frequent cluster. For example, given the question "When did Brexit happen?", if the most confident answer is $\tilde{z}_1 =$ "2020" and other unique responses include "2016" and "2019", we would prefer to include "2019" first due to its temporal proximity, even if "2016" appears more frequently. Notice that while conformal prediction for probabilistic

---

[5]For simplicity, we omit the superscript when the dependence on $q$ is clear.

classification typically constructs the predictive set incrementally by adding classes in the reverse order of their predicted membership probabilities, we find that this approach results in a predictive set that is difficult to distinguish from the remaining answers, making the task of rewriting a less specific claim unnecessarily challenging.

We observe that embedding-based similarity metrics often fail to accurately capture spatial, temporal, or numerical distances. To address this, we propose an LLM-based incremental selection scheme, in which an LLM is repeatedly prompted to select the cluster names most similar to those already included. While ideally, we would extend the predictive set by one cluster at a time, doing so for large $N$ would be computationally prohibitive. Instead, we begin with a predefined set of target thresholds $\lambda_1, \lambda_2, \ldots, \lambda_t$ and, given a subset $\tilde{Z}_n$ already selected, we prompt the LLM to select an additional $d$ items, where

$$d = \min_{\lambda \in \Lambda} \left\lceil \left( \lambda - \mathrm{Mult}(\tilde{Z}_n) \right) \frac{|Z \setminus \tilde{Z}_n|}{\mathrm{Mult}(Z \setminus \tilde{Z}_n)} \right\rceil, \quad \text{s.t. } \lambda - \mathrm{Mult}(\tilde{Z}_n) > 0,$$

where $\mathrm{Mult}(Z) = \sum_{z \in Z} m(z)$. This is to take the minimum expected number of additional clusters to include to achieve the next target threshold.

**Belief Probing**   The final step in the claim rewriting process involves associating a more general claim with each nested cluster set $\tilde{Z}_n$ using a rewriting function $f : 2^{\mathcal{Y}} \to \mathcal{Y}$. To accomplish this, we verbalize both $\tilde{Z}_n$ and its complement $\Omega \setminus \tilde{Z}_n$ as the beliefs of a hypothesized question-answering agent. We then prompt the LLM to summarize this belief in a less specific claim $b_n$ that arims to satisfy Equation (2).

We find that explicitly framing these clusters as the belief of an error-prone agent—rather than as objective facts—is crucial (See Table 3 in Appendix A). Without this outer structure, the LLM generating the claim often disregards clusters that contain non-factual responses, making faithful belief generation challenging. This belief summarization process helps the model adhere to the coherence theory of truth [72]. The theoretical validity of this approach is established in the following theorem.

**Theorem 4.2.** *For claims $b_1$ that describe $V_1$ and $b_2$ that describes $V_2$, if $b_1 \to b_2$ then*

$$V_1 \subseteq V_2.$$

The proof follows directly from the argument presented in Appendix B. While our approach performs well in practice, the unconstrained nature of the prompting mechanism means that it does not inherently guarantee $f(\tilde{Z}_j) \to f(\tilde{Z}_i)$ for all $j > i$. This limitation further justifies our choice to calibrate using the Learn-Then-Test (LTT) framework [1] rather than a simpler method like Conformal Risk Control [2]. Still the validity of the summaries has been confirmed by a small-scale manual verification conducted by one of the authors. We sampled 11 instances from each of five question types of SimpleQA [74] ("Date," "Number," "Place," "Person," "Other") and asked a human annotator to identify all answers entailed by the LLM-generated belief. Two "Person" questions were excluded due to vague criteria involving subjective judgments (e.g., whether a name belongs to a famous chemist or is a "traditional" surname). On the remaining examples, annotation precision and recall were relatively high: 0.81 and 0.91, respectively.

## 4.2   Conformalizing Rewriting with LTT

After generating a sequence of progressively less precise claims starting from the most frequent answer cluster for each question, we apply Learn-Then-Test (LTT) [1] to linguistically calibrate the response, ensuring it aligns with the optimal specificity level while maintaining the desired factuality guarantee.

**Risk Score**   In theory, the expectation of any loss function $l$, where the risk is defined as $R(F_\lambda \circ \mathcal{L}) := \mathbb{E}[l(F \circ \mathcal{L}(x), Y)]$, satisfies the requirements for LTT.[6] However, for a controlled comparison, we specifically focus on factuality metrics that do not penalize generality. Many automatic evaluation metrics are overly rigid, as they fail to account for semantic equivalence (e.g., exact match) or reject

---

[6]We use ∘ to denote function composition.

claims that differ in specificity from the gold target [43, 74]. In LLM-based evaluations, this issue can often be mitigated by slight modifications to the evaluation prompt, as detailed in Appendix A. to yield to following loss function:

$$l(\hat{y}, y) = \begin{cases} 1, & \text{if } \hat{y} \text{ is admissable} \\ 0, & \text{otherwise} \end{cases},$$

In this setting, we naturally use the multiplicity threshold as the hyperparameter $\lambda$. Specifically

$$\mathcal{T}_\lambda(\tilde{z}_1) \coloneqq f(\tilde{Z}_{n'}),$$

$$\text{where } n' = \arg\min_n \text{ Multi}(\tilde{Z}_n) \quad \text{s.t. Multi}(\tilde{Z}_n) \geq \lambda.$$

Notice that due to the discrete natural of $M$, on the same question $q$, different $\lambda$ might lead to the model generalize the original claim to the same vaguer claim. Then given $\alpha$ the goal is to find valid $\lambda$ such that

$$\mathbb{P}\Big(R(\mathcal{T}_\lambda \circ \mathcal{L}) \leq \epsilon\Big) \geq 1 - \alpha,$$

with regard to risk-tolerance $\delta$ and error level $\alpha$. This is called by Angelopoulos et al. [1] as an $(\alpha, \delta)$-risk-controlling-prediction (RCP). Following Bates et al. [4], we calculate Hoeffding-Bentkus inequality $p$-values.

To summarize, for a given claim $c$ and a designated risk control level $\alpha$, our method finds $\lambda_\alpha$ and corresponding $\mathcal{T}_\lambda(c)$ to form a four element tuple $\big(c, \alpha, \lambda_\alpha, \mathcal{T}_\lambda(c)\big)$.

## 5 Experiments and Results

We conduct three sets of experiments to validate key claims in the paper. Section 5.1 shows that by carefully controlling the LLMs to respond less specifically, we can greatly improve their factuality and maintain valid guarantee over factual errors. Section 5.2 further demonstrates that this specificity-decreasing rewriting indeed reduces new information provided by responses from LLM responses.

Since the CLC process involves oversampling and multiple rounds of LLM processing—which can be computationally expensive—we also explore, in a different domain, whether this procedure can be distilled into the LLM itself to enable efficient, high-quality rewrites, as shown in Section 5.3.

### 5.1 Linguistically Calibrated Question Answering

**Dataset** We apply the Conformal Linguistic Calibration pipeline described in Section 4 to two Question Answering (QA) tasks, and examine how gradually less specific rewritings affect the factuality of the LLM response: **SimpleQA [74]** is a recent benchmark designed to evaluate language models' ability to answer short, fact-seeking questions. Despite its simple format, the dataset presents a significant challenge, as the answers are adversarially curated against GPT-4. The authors demonstrate that even state-of-the-art models like GPT-4o perform poorly on this benchmark, achieving only 38.2% accuracy while exhibiting overconfident predictions. For our experiments, we create calibration set and test set making sure that each answer type is split evenly. **Natural Question [33]** is a widely studied QA dataset with short or long answers paired with real search queries issued to the Google search engine. To better align with the SimpleQA setting, we focus on evaluating on a subset of NQ *dev* that admits short answers that are not yes or no. As NQ can have more than one human annotation for the correct short answer, we consider an LLM response factual if it is evaluated as consistent against any one of the human-extracted correct answers.

**Evaluation Metric** While traditional QAs are typically evaluated with exact matches (EM) or semantic equivalence [79], as discussed in Section 4.2 we need a coverage metric that does not unfairly penalize responses that are less specific than the gold answers, as detailed in Appendix A. We thus modify the evaluation prompt from SimpleQA [74] to evaluate whether the less specific response from the model is relevant and consistent with corresponding gold answer, similar to factuality evaluation [43, 64]. We use Llama3-8B-Instruct [15] as base model $\mathcal{L}$, generating $K = 100$ responses per question. We then produce progressively less precise claims as described in Section 4.1, using GPT-4o while targeting multiplicities $\tau \in \{20, 30, 40, 50, 60, 70, 80, 90, 100\}$. Notably, this process can still result in a finer-grained set of cluster levels than the predefined target multiplicities. Consequently, during the LTT stage, the controlling multiplicity threshold parameter $\lambda$ is any integer within $(0, 100]$.

**Baseline**   We compare our approach against a standard abstention baseline, where the model may choose not to answer if its confidence is below a certain threshold. Under this setting, we define the *percentage* characterizing how much the model back off from the original generation as the percentage of questions the model abstain from answering, and we compare performance with our approaches at the same marginal answer set coverage percentage. That is, an abstention model that answers $p$ percent of questions are compared against a CLC model that responds with more generic answers at the level that on average covers $p$ percent of the sampled answer set. Notice that As $p \to 1$, both approaches provide little information, and as $p \to 0$ both approaches converge to not modifying original answers at all. To make sure the comparison is fair, for each question we use the same set of 100 sampled responses to quantify answer uncertainty for both approaches, and use the multiplicity of the top-1 answer as the confidence score. Since the answer formats in both datasets are short and atomic—admitting no further decomposition [72]—this setup, in the limit, aligns with the single-claim generation setting described by Mohri and Hashimoto [44]. The corresponding results are shown in the left and middle plots of Figure 3.

Our results show that for both SimpleQA (Figure 3 **Left**) and NQ (Figure 3 **Middle**), the factuality of responses from our approach tightly aligns with the lower bound of the conformal guarantee region, indicating it avoids generating trivially true answers. Compared to traditional abstention, our method offers three advantages. First, while abstention improves factuality by answering only a small subset of questions, CLC answers all questions while still enhancing factuality. Second, on challenging datasets like SimpleQA—where models struggle to confidently answer even a few questions—our method significantly outperforms abstention. Third, CLC yields more consistent factuality gains as the relaxation percentage increases, whereas abstention becomes unstable at higher abstention ratios.

## 5.2   Informativeness Scoring

We conduct further evaluation to show that the improvements in factuality comes in trade-off with less specificity. To achieve this, we evaluate the **informativeness** of our generation at different answer multiplicity. We adopt the Conditional Pointwise Mutual Information (CPMI) [27] as the informativeness measure, which quantifies informativeness as the negative log likelihood of a generation given what was already known in the prompt. We operationalize CPMI with a publically-available general conditional probability estimator [68].[7] To make sure that the conditional probability is not confounded by the parametric knowledge, we call Qwen2.5-72B-instruct to mask personal entity with "PersonX", similar to how ATOMIC commonsense reasoning data is created [60]. We also compare our approach to a **naive** rewriting baseline where the base model (Llama-3-7b-Instruct) is prompted[8] to directly rewrite the response to boost confidence to at least equal to the multiplicity.

The result is shown in Figure 4. For both SimpleQA and NQ, while the factuality of the responses increase with growing multiplicity threshold, their CPMI steadily decrease. Our results demonstrate that CLC enables explicit factuality-informativeness trade-off. What's more, the naive rewriting baseline failed to facilitate fine-grained trade-off, and easily degenerates to rewrites with low informativeness.

## 5.3   Claim Rewriting on FActScore

Since the original CLC process is costly as it requires answer over sampling and a pipeline for claim rewriting, we further show that this process can be amortized. To this end, we fine-tune Llama-3-8B-Instruct [15] on the synthetic data generated in Section 5.1, which consists of 2,042 instances from SimpleQA and 1,728 instances from NQ with various level of back-off generation. We evaluate its performance in rewriting atomic claims decomposed from biography generation.

Specifically, we assess changes in FActScore [43] before and after applying the claim rewriter $\mathcal{T}$ to all decomposed atomic sub-claims. FActScore is a method that evaluate the precision of supported claims decomposed from a generation. For evaluation, we use the *dev* split of the Core [27] expansion of the FActScore dataset, which consists of an extended list of biography topics beyond those in the original FActScore dataset and includes frequency values associated with each topic.

---

[7]`https://huggingface.co/Zhengping/conditional-probability-regression`
[8]As shown in Table 5.

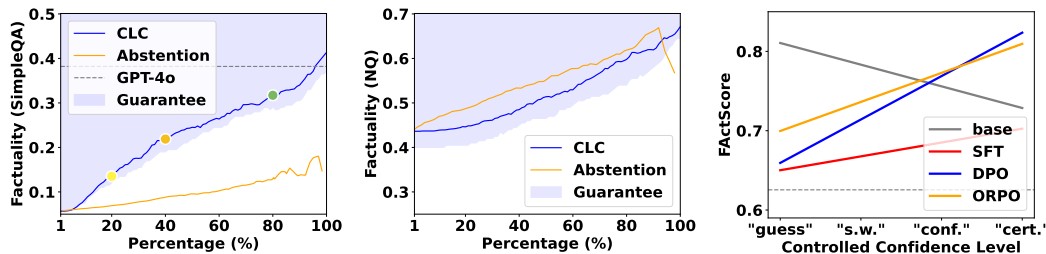

Figure 3: **Left**: On SimpleQA, the prediction risk on the test set falls within the conformal guarantee region and remains closely aligned with the respective lower accuracy thresholds, while outperforms abstention by a large margin.[10] We sample answers to the same question, "Pepi Litman was born in …", at different multiplicity thresholds: (⬛ "Kiev / Kyiv." ✗), (⬛ "a major cultural and historical center in central or eastern Ukraine." ✗), (⬛ "a major cultural and historical center of Ukraine." ✓). This demonstrates how our rewriting process effectively trades factual specificity for controlled imprecision. **Middle**: On datasets where the model already achieves non-trivial performance, such as Natural Questions, our guarantee remains valid, and the model still benefits significantly from adjusting precision levels. Abstention does not steadily improves factuality. **Right**: Fine-tuning an LLM on the imprecise claim rewriting data enables adaptive rewriting. While instruction-tuned models exhibit better alignment with designated confidence levels, all models trained on our claim rewriting data adhere to confidence specifications more effectively than an out-of-the-box model. Dash line indicates the out-of-the-box performance without CLC.

Given the series of weaknesses identified in the standard FActScore configuration[11], we restrict verification to Core-filtered [27] subclaims. These subclaims are assessed against BM25 [57]-retrieved text chunks from a full Wikipedia dump.

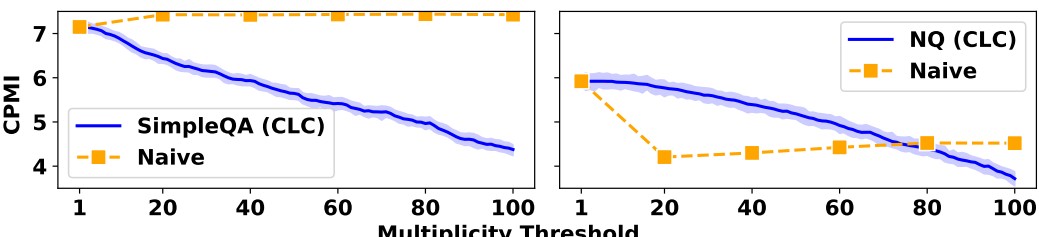

Figure 4: Average CPMI scoring of claims for **SimpleQA (left)** and **NQ (right)**. CLC rewriting results in smooth and steady decrease in CPMI scores while naive rewriting does not. Shades corresponds to 95% confidence interval.

We compare the following four claim rewriters trained on 2 A100 80G:

$\mathcal{T}_{\textbf{Base}}$: This rewriter directly prompts the off-the-shelf Llama3 8B model with the instruction: "Rewrite the following claim to be less specific until you `verbalization` it is true: `claim`."

$\mathcal{T}_{\textbf{SFT}}$: This model is fine-tuned using the same prompt on triplets of the form (`verbalization`, `claim`, `rewrite-at-`$\lambda$) derived from the SimpleQA and Natural Questions rewriting data. The threshold $\lambda$ for each verbalized uncertainty level is determined based on an analysis of the conformal guarantee presented in Figure 3.

$\mathcal{T}_{\textbf{DPO}}$: Following best practices for Direct Preference Optimization (DPO) [52, 51, 54], we further fine-tune $\mathcal{T}_{\text{SFT}}$ on preference data. Specifically, we always prefer rewrite-at-$\lambda$ over alternative rewritings that exhibit noticeable deviations.

---

[11]Including susceptibility to trivial claims, inability to verify claims involving secondary entities [75, 62, 81].

$\mathcal{T}_{\mathbf{ORPO}}$: Odds Ratio Preference Optimization (ORPO) [25] offers an alternative alignment approach by applying a minor penalty to disfavored generation styles at the SFT stage. We train this model using the same dataset as $\mathcal{T}_{\mathrm{DPO}}$.

The FActScore of the rewritten generations is presented in the right panel of Figure 3. The transition from "guess" through "somewhat confident" and "confident" to "certain" indicates the gradual raising of the required confidence level.[12] By generalizing individual claims, all of our claim rewriters effectively enhance the FActScore of the generated outputs. In contrast, the out-of-the-box Llama model ($\mathcal{T}_{\mathrm{Base}}$) does not consistently follow the instruction to rewrite claims to the designated uncertainty level. It is also interesting that the tuned model seems to learn to adapt its level of rewriting specificity based on the difficulty of the claim, as shown in Appendix C.

## 6 Conclusion

We propose a novel linguistic calibration paradigm that enables models to express and communicate uncertainty through imprecise language. Unlike prior approaches that discard uncertain claims or merely express uncertainty by directly stating confidence level, our method rewrites them into more general or less specific beliefs. Our results show that, unlike prior uncertainty-aware risk control methods, CLC enables models to respond unambiguously to all questions while steadily and effectively improving factuality and maintaining valid risk control guarantees.

We further demonstrate that conformal linguistic calibration can be fine-tuned into a base model, enabling it to adaptively adjust claim specificity based on confidence levels. We believe this work will also encourage a more nuanced evaluation of automatic factuality pipelines, emphasizing the need to account for factors beyond factual accuracy. Future work will expand this analysis to examine how AI confidence communication shapes human decision-making while acknowledging the challenges of conducting human-based factuality trade-off evaluations due to known biases [37, 14].

## 7 Acknowledgements

AL and BVD are partially supported by ONR grant N00014-241-2089.

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

# A  Prompt Templates

Table 1: Prompt for identifying unique answer to a given question.

| Field | Content |
|---|---|
| Examples | [User] Given a list of answers to a question, identify semantically distinct answers.

**List of Answers**:
${Answers}
[Agent] **Semantically Distinct Answers**:${SD_Answers} |
| Input | [User] Given a list of answers to a question, identify semantically distinct answers.

**List of Answers**:
${Answers} |

Table 2: Prompt using iterative clustering fornested set construction.

| Field | Content |
|---|---|
| Instruction | Given a list of already **selected items**, your task is to find **K additional items** from the list of **candidates** that are **most similar** to the items already selected. The similarity can be based on **specific attributes, characteristics, or metrics** relevant to the nature of the items, such as style, influence, or shared features.

You should return a list of **K items** from the **candidates** list and store them in a variable named **increments**.

—

**Inputs:**
1. **K**: The number of additional items to add to the selected list.
2. **selected**: A list of already selected items (could be names, objects, etc.).
3. **candidates**: A list of candidate items from which we need to select the K most similar items.

**For Example:**

```python
K = 1
selected = ["Red"]
candidates = ["Yellow", "Black", "White"]
```

—

**Expected Output:** Return a list of **K items** from the **candidates** list that are **most similar** to the items in the **selected** list.

```python
increments = ["Yellow"]
``` |

| | |
|---|---|
| **Examples** | ````python`
`K = 1`
`selected = ["William Butler Yeats"]`
`candidates = ["Agatha Christie", "Benjamin Franklin",`
`"Napoléon Bonaparte"]`
``` ``` `

[Agent] Based on your inputs, we are tasked with selecting 1 item from the list of candidates that is most similar to the selected item, **"William Butler Yeats"**. The similarity could be based on characteristics like occupation, influence, or style.

- **William Butler Yeats** was an Irish poet and one of the foremost figures of 20th-century literature.

- **Candidates**:
- **Benjamin Franklin**: American polymath, writer, scientist, diplomat.
- **Napoleon**: French military leader and emperor.
- **Agatha Christie**: British writer known for her detective novels.

In this case, based on occupation (both are writers), **Agatha Christie** is most similar to **William Butler Yeats**.

Thus, the result would be:

````python`
`increments = ["Agatha Christie"]`
``` ``` ` |
| **Input** | ````python`
`K = $K`
`selected = $SELECTED`
`candidates = $CANDIDATES`
``` ``` ` |

Table 3: Prompt for generating belief that describes a belief.

| Field | Content |
|---|---|
| **Instruction** | [User] Suppose a human subject is going to respond to a question but they don't know the exact answer. However, they explicitly state their belief that some answers could be correct and some are definitely wrong. Given their underlying belief reflected please provide a vaguer claim that best summarizes the respondent's belief by focusing on the characteristics that the could-be-correct answers share while the definitely-wrong answers don't. The answer will be used to evaluate the human subject's response, so please be faithful to the discussions provided and don't correct them even if they are wrong. Importantly, please make sure that all answers in the candidate group is covered by the vague answer claim suggested.

Questions and beliefs are provided below. Please respond after reasoning about the similarities and differences between the positive and negative answers. Wrap your answer in a code block (```answer```). The answer should start with "The respondent believes that". |

| | |
|---|---|
| **Examples** | [User] **Question:** Who's the best football player of all time? **Belief:** The respondent believes that the answer is either Pele, Diego Maradona, or Lionel Messi, but not Cristiano Ronaldo, Zinedine Zidane, nor Johan Cruyff.

[Agent] **Positive Group: Pelé, Diego Maradona, Lionel Messi**
*Time Period*: Spanning from the 1950s to the present.
*Geographic Origin*: South America (Brazil and Argentina).
*Influence*: Known for their extraordinary dribbling skills, creativity, and playmaking abilities.
*Philosophy*: These players are celebrated for their flair, technical prowess, and ability to change the course of a game single-handedly. Pelé (Brazil) is renowned for his goal-scoring and three World Cup victories. Maradona (Argentina) is famous for his "Hand of God" goal and his incredible dribbling, particularly in the 1986 World Cup. Messi (Argentina) is known for his consistency, vision, and record-breaking achievements with FC Barcelona and Argentina.
*Distinct Features*: Emphasis on individual brilliance, creativity, and a deep connection with their home countries' footballing culture.

**Negative Group: Cristiano Ronaldo, Zinedine Zidane, Johan Cruyff**
*Time Period*: Spanning from the 1970s to the present.
*Geographic Origin*: Europe (Portugal, France, Netherlands).
*Influence*: Known for their versatility, tactical intelligence, and leadership on the field.
*Philosophy*: These players are celebrated for their physical prowess, strategic thinking, and ability to perform in crucial moments. Ronaldo (Portugal) is known for his athleticism, goal-scoring, and adaptability across leagues. Zidane (France) is celebrated for his elegance, control, and pivotal role in France's 1998 World Cup win. Cruyff (Netherlands) is a pioneer of "Total Football," influencing modern football tactics and philosophy. *Distinct Features*: Emphasis on tactical intelligence, versatility, and significant contributions to both club and national team success.
**Key Differentiation**: The positive group is characterized by South American flair, individual brilliance, and a deep cultural impact on football. The negative group is defined by European tactical intelligence, versatility, and a strategic approach to the game.

```
The respondent believes that the best football player
of all time comes from South America.
```
|
| **Input** | [User] **Question**: ${question}

**Belief**: ${belief} |

Table 4: Prompt for evaluating factuality when imprecision occurs on SimpleQA [75].

| Field | Content |
|---|---|
| **Instruction** | [User] Your job is to look at a question, a gold target, and a predicted answer, and then assign a grade of either ["CONSISTENT", "INCONSISTENT", "NOT_ATTEMPTED"].
First, I will give examples of each grade, and then you will grade a new example. |

**Examples** The following are examples of CONSISTENT predicted answers.
```

Question: What are the names of Barack Obama's children?
Gold target: Malia Obama and Sasha Obama
Predicted answer 1: sasha and malia obama
Predicted answer 2: Each of them has a girl's name.
Predicted answer 3: They have the same last name as their father, Obama.
```

These predicted answers are all CONSISTENT because:
- They do not contain any information that contradicts the gold target.
- Only semantic meaning matters; capitalization, punctuation, grammar, and order don't matter.
- Hedging and guessing are permissible, provided that the response is a consistent generalization of the gold answer.

The following are examples of INCONSISTENT predicted answers.
```

Question: What are the names of Barack Obama's children?
Gold target: Malia and Sasha
Predicted answer 1: Malia.
Predicted answer 2: Barack Obama does not have any children.
Predicted answer 3: It's possible you may mean Betsy and Olivia. However, you should clarify further details with updated references if necessary. Is that the correct answer?
Predicted answer 4: It may be the case that Obama's child is named James. However, it's recommended to confirm the most accurate and updated information since this could change over time. This model may not always reflect the most current information.
```

These predicted answers are all INCONSISTENT because:
- An inconsistent statement in the answer that contradicts the gold target. Inconsistent statements that have some hedging (e.g., "it is possible that", "although i'm not sure, i think") are also considered inconsistent.
- Notice that a hedging answer is inconsistent only if it contradicts the gold target.

The following are examples of NOT_ATTEMPTED predicted answers.
```

Question: What are the names of Barack Obama's children?
Gold target: Malia and Sasha
Predicted answer 1: I don't know.
Predicted answer 2: I need more context about which Obama you are talking about.
```

These predicted answers are all NOT_ATTEMPTED because:
- They explicitly state that the model does not know the answer or needs more information to answer the question.

Also note the following things:
- Do not punish predicted answers if they omit information that would be clearly inferred from the question.
- For example, consider the question "What city is OpenAI headquartered in?" and the gold target "San Francisco, California". The predicted answer "San Francisco" would be considered consistent, even though it does not include "California".
- Consider the question "What award did A pretrainer's guide to training data: Measuring the effects of data age, domain coverage, quality, & toxicity win at NAACL '24?", the gold target is "Outstanding Paper Award". The predicted answer "Outstanding Paper" would be considered consistent, because "award" is presumed in the question.

| Field | Content |
|---|---|
| **Examples** | - For the question "What is the height of Jason Wei in meters?", the gold target is "1.73 m". The predicted answer "1.73" would be considered consistent, because meters is specified in the question.
- For the question "What is the name of Barack Obama's wife?", the gold target is "Michelle Obama". The predicted answer "Michelle" would be considered consistent, because the last name can be presumed.
- Do not punish for typos in people's name if it's clearly the same name.
- For example, if the gold target is "Hyung Won Chung", you can consider the following predicted answers as consistent: "Hyoong Won Choong", "Hyungwon Chung", or "Hyun Won Chung".
- A response being too general is not a reason to mark it as INCONSISTENT.
- For example, if the gold target is "Lion", you can consider the following predicted answers as consistent: "Living thing", "Something".
- For the question "Who wrote the book 'The Great Gatsby'?", the gold target is "F. Scott Fitzgerald". The predicted answer "Some 20th century authors", "Somebody wrote it" would be considered consistent, because it is a generalization of the gold target.
- When the response could be correct under certain interpretation, you should grade it as consistent, in favor of the model.
- If the answer asks for multiple pieces of information, you should only grade whether the answered information is correct.
- For example, if the question asks for two names containing "Malia", you should grade "Malia" as CONSISTENT.
- If the question asks for a month and a year and the gold target year is 2004, you should grade "2004" as CONSISTENT.
- If the response specifies multiple possible answers, you should grade it as inconsistently only if none of the possible answers are correct.
- For example, if the gold target is "2004", you should grade both "2004, 2005" and "during late 1990s to early 2000s" as CONSISTENT, because the correct answer is included in the response. |
| **Input** | `[User] ```` 
Question: `${question}`
Gold target: `${target}`
Predicted answer: `${predicted_answer}`
` ``` ` |

Table 5: Claim rewriting template. The verbalization is associated to the designated level. For the levels qualitatively examined in the paper, **Guess** corresponds to the verbalization "would guess", **S. Conf.** corresponds to "are somewhat confident", and **Conf.** corresponds to "are confident".

| Field | Content |
|---|---|
| **Input** | `[User]` the following claims to be less specific until you `${verbalization}` is is true: `${source_claim}`

Your response should only contain the claim itself, without any additional context. |

# B   Nested Property

*Proof of Theorem 4.2.* Suppose there's a possible world $v \in V_1$, but $v \notin V_2$. Given that

$$(b_1 \rightarrow b_2) \rightarrow (\Box(b_1 \rightarrow b_2))$$

Thus by Definition 3.1 and Definition 4.1 we have $v \Vdash b_2$. However, since $v \notin V_2$ we have $v \in W \setminus V_2$, again by Definition 4.1 we have $v \Vdash \neg b_2$, which is a contradiction. $\qquad\square$

## C    More Examples

In this section of the appendix, we examine additional examples to illustrate how our trained claim rewriter adaptively modifies given claims as shown in Table 6, shifting them toward varying levels of uncertainty. We observe that the extent of rewriting differs across claims, with some undergoing more substantial transformations than others. Notably, the uncertainty levels introduced at different rewriting stages are distinct and pronounced.

## D    System Evaluation

We conduct additional evaluation to critical components in our system to make sure that they are robust enough for the main conclusion to hold.

### D.1    Number of Answer Samples Needed

We study how many sampled answers are required for our clustering pipeline to recover the diversity of plausible completions. For each question we run the answer sampling and iterative clustering routine described in Section 4 while capping the sampling budget at $K \in \{20, 40, 60, 100\}$. We then record the number of unique clusters that survive the pruning step and average across 500 held-out prompts. The resulting curve, summarized in Table 7, shows diminishing returns: quadrupling the budget from 20 to 80 samples (interpolated between reported points) yields only moderate gains in cluster coverage. While Learn-Then-Test (LTT) remains valid even when $K = 20$, we observe that higher $K$ provides more nuanced uncertainty control, but even K=20 maintains sufficient cluster diversity for effective calibration.

### D.2    Belief Probing Robustness

To gauge how the belief probing stage behaves in practice, one author manually inspected 50 sampled prompts across different SimpleQA question types. Each inspection checked whether the model-authored belief summary entailed answers in the set it needs to summarize. The review achieved precision 0.81 and recall 0.91, suggesting that the summarization process reliably captures the intended answer sets and that occasional mis-clusterings do not materially weaken our risk guarantees.

## E    Impact Statement

Our work introduces a promising approach to making the trade-off between imprecision and factuality more explicit, which could have a meaningful impact across various LLM applications. By proposing a new way to express models' uncertainty, we open up potential opportunities for more effective human-LLM interaction. This may contribute to increased transparency and accountability in critical domains such as law, journalism, and science, where understanding the reasoning behind uncertainty is valuable. Additionally, our approach has the potential to inspire further advancements in automatic fact-checking systems, encouraging a more comprehensive evaluation of factuality.

## F    License and Agreement

The SimpleQA dataset [74] is released under the MIT license. We use the version released with the simple-eval repository.[13] The Natural Question dataset [33] is released under the Apache-2.0 license. We use the open subset.[14] The open-weight model we use includes models from the Llama3 family, which are released under the Llama3 license, and models from or fine-tuned from the Qwen2.5 family, which are released under Apache-2.0 license.

---

[13]https://github.com/openai/simple-evals
[14]https://github.com/google-research-datasets/natural-questions.git

Table 6: More examples occurring when backing off claims from the CORE dataset.

| Claim |
| --- |
| *"Lost in White" is a horror-thriller.* |
| **Guess**⤳ "Lost in White" is a horror-thriller. |
| **S. Conf.**⤳ "Lost in White" is a type of film that blends elements of horror and thriller genres. |
| **Conf.**⤳ "Lost in White" is a film that blends elements of thriller and horror genres. |
| *Carl Rackemann played 10 One-Day Internationals (ODIs).* |
| **Guess**⤳ Carl Rackemann played 10 One-Day Internationals (ODIs). |
| **S. Conf.**⤳ the number of One-Day Internationals played by Carl Rackemann could be any positive integer from 0 to 143. |
| **Conf.**⤳ the number of One-Day Internationals played by Carl Rackemann is a positive integer. |
| *Kadegaon is located in the Kolhapur district.* |
| **Guess**⤳ Kadegaon is located in the Kolhapur district. |
| **S. Conf.**⤳ Kadegaon is located in the Kolhapur district. |
| **Conf.**⤳ the village Kadegaon is located in a district of Maharashtra, India. |
| *Tim Whitnall was born on May 12, 1954.* |
| **Guess**⤳ Tim Whitnall was born on the 12th of a month in 1954. |
| **S. Conf.**⤳ Tim Whitnall was born on a day that is any date within a wide range of possible dates, or they are unsure of his birthdate. |
| **Conf.**⤳ there is no available information about Tim Whitnall's birthdate, and they are uncertain or unaware of it. |
| *Yoon Seok-ho's films explore themes of trauma.* |
| **Guess**⤳ Yoon Seok-ho's films explore themes of trauma. |
| **S. Conf.**⤳ Yoon Seok-ho's films explore themes related to human emotional experience and social issues. |
| **Conf.**⤳ Yoon Seok-ho's films explore themes of human emotional experience and social commentary. |
| *Tadashi Imai died in 1999.* |
| **Guess**⤳ Tadashi Imai died in 1999. |
| **S. Conf.**⤳ Tadashi Imai died sometime between the 1960s and 2000. |
| **Conf.**⤳ the date of Tadashi Imai's death is uncertain or not specifically known. |
| *Ernie Dingo gained national recognition for his role as Jacko in the television series "The Sullivans."* |
| **Guess**⤳ Ernie Dingo gained national recognition for his role as Jacko in the television series "The Sullivans." |
| **S. Conf.**⤳ there is a possibility that Ernie Dingo gained national recognition for his role as Jacko in "The Sullivans," but they are uncertain about the specific year or character name. |
| **Conf.**⤳ the actor who gained national recognition for his role as Jacko in "The Sullivans" is an Australian entertainer from the mid to late 20th century. |
| *Zeki Demirkubuz's films feature non-linear narratives.* |
| **Guess**⤳ Zeki Demirkubuz's films feature non-linear narratives. |
| **S. Conf.**⤳ Zeki Demirkubuz's films feature complex narrative structures that explore themes of identity and human experience. |
| **Conf.**⤳ Zeki Demirkubuz's films feature complex and unconventional narrative structures. |
| *Zia Fatehabadi used the "radif" in his ghazals.* |
| **Guess**⤳ Zia Fatehabadi used the "radif" in his ghazals. |
| **S. Conf.**⤳ the use of "radif" in Zia Fatehabadi's ghazals is a traditional and structural element of classical Urdu poetry. |
| **Conf.**⤳ the use of "radif" in ghazals is a traditional feature of classical Urdu poetry. |

Table 7: Average number of clusters while sampling budget $K$ increases.

| $K$ | Avg. # Cluster |
| --- | --- |
| 20 | 8.676 |
| 40 | 13.735 |
| 60 | 17.806 |
| 100 | 24.461 |

