# OpenReview forum: "Conformal Linguistic Calibration: Trading-off between Factuality and Specificity"
_NeurIPS.cc/2025/Conference — NeurIPS 2025 poster_

### Official Review · Reviewer_ayvU · 2025-06-30

**Clarity:** 2
**Significance:** 2
**Originality:** 3
**Rating:** 4
**Confidence:** 3

**Summary:**

This paper introduces Conformal Linguistic Calibration (CLC), a framework that turns the intuitive act of “hedging” with linguistic qualifiers into a formal, distribution-free procedure with factual correctness $\geq 1 - \alpha$ while retaining as much specificity as possible.
The pipeline (i) samples multiple candidate answers from a base LLM, (ii) groups them into semantic clusters via a single additional LLM call, (iii) orders the clusters to form a nested sequence of answer sets, (iv) asks the LLM to rewrite each set into a single “belief summary” sentence, and (v) uses Learn-Then-Test conformal calibration to choose the most specific rewrite whose empirical error is below $\epsilon$ at level $\alpha$. At deployment time the system outputs that rewrite, trading specificity for coverage. Experiments on NQ and SimpleQA show that CLC provides the promised error guarantees and retains substantially more information than an abstention-based baseline.

**Questions:**

- Authors mention that they use an LLM to generate a list of unique answer cluster names from {a} (instead using more standard entailment models to establish equivalency relationships). Is the performance of LLM clustering analyzed or discussed? How robust is the system to the cluster generation process?
- Please further clarify how semantic equivalency is determined in “...answers that are semantically equivalent to the corresponding cluster name...”?
- “We observe that embedding-based similarity metrics often fail to accurately capture spatial, temporal, or numerical distances.“ --- Were embedding-based similarity metrics tried? Please provide comparison results if so.

**Ethical Concerns:**

["NO or VERY MINOR ethics concerns only"]

**Final Justification:**

My specific questions have been adequately addressed by the authors' rebuttal, but the initially outlined limitations remain. I think this is promising work, but would still need to see additional experiments to properly contextualize these results; therefore I maintain the borderline accept suggestion.

**Limitations:**

Limitations are partially discussed in Section 3.1; but limitations of the pipeline brittleness or compute (API calls) budget should be discussed more in-depth.

**Paper Formatting Concerns:**

Misspellings or confusing sentences:
- L. 83 & L.91: "accesibility"
- L. 106: "gurantees"
- L. 192: "that _arims_ to satisfy"
- L. 223-224: "admissable"
- L. 225: "due to the discrete _natural_ of M"

Also, re-check Figure 2 caption formatting.

**Quality:**

3

**Strengths And Weaknesses:**

**Strengths**
- Clearly motivates the need for partial rather than binary abstention and formalises it with conformal guarantees.
- Careful empirical evaluation on two QA datasets; conforms to $\alpha$-risk.
- Fine-grained CPMI analysis; and experiments show promising results.

**Weaknesses**
- Heavy reliance on heuristic LLM clustering; no quantitative study of clustering accuracy or ablations on sphere-construction choices.
- Baseline comparisons limited to a simple abstention baseline; perhaps could be compared to a baseline embedding-similarity or entailment-based set prediction (at least parts of the CLC pipeline could be compared to baselines in isolation).
- Ablations or robustness of each stage of the pipeline is not really assessed.
- Computational cost is not fully benchmarked, but it appears to be much higher than related uncertainty quantification methods.
- Other potential weaknesses below in the "questions" section.

---
- It would be a great addition if you could wrap your contributions in an easy to use python package that could be plugged into open-source models or a web API.

---

> ### Author Rebuttal · Authors · 2025-07-31
>
> Thank you for finding our partial abstention motivation compelling and appreciating our conformal guarantee formalization. We address your concerns about component robustness and computational costs below.
>
> ## Q1: Clustering Component Evaluation
>
> While comprehensive evaluation is challenging without ground truth labels, we optimized each pipeline component through careful design choices:
>
> **Identifying Clusters:** We initially tested entailment-based approaches but found NLI models overly sensitive to surface forms and noisy (McCoy et al., 2019). Embedding-based approaches work poorly for numeric cases (Naik et al., 2019; Tang et al., 2025). For example, in one of our test cases “2023” is more similar to “2026” than “2024” in embedding space. Since we can’t predict if an LLM output will be numeric, number-specific methods are also hard to apply.
>
> **Estimating Answer Multiplicity (How semantic equivalence is determined):** We use embedding-based similarity (answer will be assigned to the most similar cluster, which we deem to be semantically equivalent) for answer multiplicity estimation (Section 3.1). Here using embedding-based similarity to identify semantic equivalence is more reliable as unique clusters have been identified and usually there will be one cluster that is very similar to the answer at hand. In this case, we only need to find the most similar cluster, not discover new ones. Previous research suggests that despite noisy clustering, uncertainty estimation from traditional clustering approaches remains reliable (Kuhn et al., 2023; Grewal et al., 2024). While clustering accuracy is hard to evaluate directly without ground truth label, it is indirectly evaluated together with our belief probing verification study described below.
>
> **Belief Probing:** One of the authors conducted manual verification on over 50 examples from different prompt categories to see whether a unique answer is entailed by the summary. The manual verification study shows **precision: 0.81, recall: 0.91** across different question types, indicating robustness.
>
> We also would like to note that we use LTT for calibration, which is based on hypothesis testing and only requires an independent and identically distributed calibration set. Even if no pipeline components are perfect, the LTT guarantee remains valid, although the performance of each component does affect the optimality of the trade-off curve.
>
> ## Q2: Computational Cost
>
> We acknowledge the computational overhead of curating high-quality answer sets. To address this, our experiments demonstrate that hedging behavior can be distilled into base models for direct generation. Other relatively expensive approaches that require sampling also resort to similar ideas (Kuhn et al., 2023; Farquhar et al., 2024; Band et al., 2024).
>
> ## Q3: Open Source and Technical Issues
>
> We will fix all formatting issues, including the last line of Figure 2, and will **open source our complete dataset and step-wise creation code with the camera-ready version.**
>
> ## References
>
> - *McCoy, R. Thomas, Ellie Pavlick, and Tal Linzen. "Right for the wrong reasons: Diagnosing syntactic heuristics in natural language inference." arXiv preprint arXiv:1902.01007 (2019).*
> - *Naik, Aakanksha, et al. "Exploring numeracy in word embeddings." Proceedings of the 57th Annual Meeting of the Association for Computational Linguistics. 2019.*
> - *Tang, Eric, Bangding Yang, and Xingyou Song. "Understanding LLM Embeddings for Regression." Transactions on Machine Learning Research.*
> - *Kuhn, Lorenz, Yarin Gal, and Sebastian Farquhar. "Kuhn, Lorenz, Yarin Gal, and Sebastian Farquhar. "Semantic Uncertainty: Linguistic Invariances for Uncertainty Estimation in Natural Language Generation." The Eleventh International Conference on Learning Representations.*
> - *Grewal, Yashvir S., Edwin V. Bonilla, and Thang D. Bui. "Improving uncertainty quantification in large language models via semantic embeddings." arXiv preprint arXiv:2410.22685 (2024).*
> - *Farquhar, Sebastian, et al. "Detecting hallucinations in large language models using semantic entropy." Nature 630.8017 (2024): 625-630.*
> - *Band, Neil, et al. "Linguistic Calibration of Long-Form Generations." International Conference on Machine Learning. PMLR, 2024.*

---

> > ### Comment · Reviewer_ayvU · 2025-08-05
> >
> > Thank you for the detailed rebuttal; several of my earlier questions are now cleared up, but several points are still unaddressed:
> > -  I appreciate the commitment to release code and data, but I don't wish to encourage the poor practice of releasing code conditioned on acceptance; code and other artifacts are important for the review.
> > - Improved ablation studies and robustness analysis: given the many moving parts of the pipeline, it would be important to have ablations over the different components (e.g., model choice, choice of K clusters, prompting strategy); Also, robustness to errors within these pipeline components (e.g., injecting controlled noise into an upstream pipeline stage and assessing how it impacts performance downstream).
> >     - It seems you probably have gone through many of these ablations during the research period; perhaps discussing these different approaches and why they didn't work would be a good addition to the appendix.
> > - I agree with reviewer `1Ph2` on "Many recent works also combine conformal prediction with abstention mechanisms for uncertainty calibration, and the paper should either compare to these methods or provide a clear argument for why a direct comparison is not meaningful."
> > - You currently have additional related work discussion in Appendix A. Unfortunately this section can't simply be moved to the appendix: improved contextualization of the work within related literature is a must in the main body of the paper (e.g., comparison with prior conformal abstention and linguistic hedging papers).
> >
> > Overall: I thank the authors for the clarifications, and will maintain my positive suggestion of (borderline) accept; The paper could be a definite accept with improved contextualization within related literature, improved baselines, and proper ablations in most stages of this complex CLC pipeline.

---

> > > ### Author Response · Authors · 2025-08-07
> > >
> > > We appreciate the reviewer's clear feedback on how we can further improve this work and will address the remaining concerns to the best of our ability.
> > >
> > > ## Ablation Study
> > >
> > > As discussed in our original rebuttal, given the substantial number of API calls required for data creation and the need for a high-quality LLM-as-a-judge in SimpleQA evaluation, we conducted ablation studies on a smaller scale rather than running the full pipeline with different configurations. Our ablation uses 190 samples from SimpleQA for calibration and 192 samples for testing, designed to closely match the answer_type distribution in the full dataset. We focused on the key pipeline design choices that reviewers were most interested in:
> > >
> > > - LLM for dataset processing (GPT-40 -> Llama-3.1-8B-Instruct)
> > > - LLM-based iterative clustering (LLM -> Embedding based approach with `paraphrase-MiniLM-L6-v2`)
> > > - Adding outer structure prompts for belief-modeling (Belief -> Direct Answer Summarization)
> > > The result is shown below:
> > >
> > > | Method                           | Threshold 0 | Threshold 20    | Threshold 40    | Threshold 60    | Threshold 80    |
> > > | -------------------------------- | ----------- | --------------- | --------------- | --------------- | --------------- |
> > > | $-$ GPT-4o                       | 0.0677 (0)  | 0.1302 (0.0550) | 0.1927 (0.1450) | 0.2604 (0.1700) | 0.3177 (0.1700) |
> > > | and $-$ LLM-iterative-clustering | 0.0628 (0)  | 0.1204 (0.0550) | 0.1789 (0.1150) | 0.2021 (0.1700) | 0.2926 (0.2100) |
> > > | and $-$ Belief-modeling prompt   | 0.0677 (0)  | 0.0625 (0)      | 0.0469 (0)      | 0.0365 (0)      | 0.0260 (0)      |
> > >
> > > The `test_accuracy` is shown with the guarantee provided by LTT in parentheses. These results are directly comparable with the left panel of Figure 3. We also have similar figures illustrating detailed results for each condition, which we can include in future versions. In general:
> > >
> > > - While the conformal guarantee still holds with LTT, using a smaller LLM considerably hurts performance.
> > > - Similarity-based clustering slightly worsens accuracy at these thresholds.
> > > - Removing the outer structure and demonstrations for belief modeling has a devastating impact on the model's capability to generate high-quality back-off claims.
> > >
> > > As the reviewer noted, our goal is to develop a functional CLC pipeline, and we have considered and preliminarily tested many of these approaches. We believe this new ablation study, combined with the annotation study presented in the previous round, comprehensively addresses the reviewer's concerns regarding pipeline implementation.
> > >
> > > ## Related Work and Baseline Choices
> > >
> > > We agree with the reviewer that discussion of related work will be very helpful in contextualizing the work, and the reason it is in appendix is mainly due to page limit. And that's why we try to provide a concise summary of how our work contrasting with existing work in the intro. With an additional page we will make sure such discussion is included in the main body of the text.
> > >
> > > As to why only one baseline is selected for comparison, we would like to maintain that our approach differs fundamentally from standard abstention methods by enabling partial abstention at the claim level. While existing methods refuse to answer entirely, we can respond with partial knowledge when confidence is relatively low (see Figure 3 example). Our baseline already compared against the closest approach (Mohri and Hashimoto, 2024) in Section 4.1, and we believe this is already sufficient in showing how our proposal is very different from standard abstention approaches. As in our response to reviewer 1Ph2, abstention is a relatively standardized practice and most works share a very similar structure to the baseline we compared to on short response QA (Yadkori et al., 2024). And from a previous study the sample-based uncertainty we adopt for our baseline is generally considered the best non-conformity score for this purpose (Cole et al., 2023). The fundamental benefit one has with CLC is that it can improve performance even when the task is very challenging where there's no subset of the questions that the model can answer precisely and correctly, which is fundamentally very difficult given the formulation of ablation.
> > >
> > > We're pleased that the reviewer found our previous discussion valuable and believe this additional information will help address the remaining concerns raised.
> > >
> > > ## References
> > >
> > > - _Mohri, Christopher, and Tatsunori Hashimoto. "Language models with conformal factuality guarantees." Proceedings of the 41st International Conference on Machine Learning. 2024._
> > > - _Yadkori, Yasin Abbasi, et al. "Mitigating llm hallucinations via conformal abstention." arXiv preprint arXiv:2405.01563 (2024)._
> > > - _Cole, Jeremy, et al. "Selectively Answering Ambiguous Questions." Proceedings of the 2023 Conference on Empirical Methods in Natural Language Processing. 2023._

---

### Official Review · Reviewer_xhfo · 2025-07-02

**Clarity:** 3
**Significance:** 3
**Originality:** 3
**Rating:** 5
**Confidence:** 3

**Summary:**

This paper proposes Conformal Linguistic Calibration, a claim-rewriting framework that reinterprets the uncertainty of the model as set prediction; to give a factual answer, the model tailors its specificity accordingly. For instance, if the confidence is high, the model will give a very specific response. If the confidence is rather low, the model will give a rather less specific response.

The method is applied to the task of Question Answering. First N answers are sampled from the LLM which are then clustered together by trying to find semantically unique clusters. The amount of resulting clusters can indicate the certainty of the LLM. The LLM then creates nested clusters which are after associated with a more general claim. The more general claims are then conformalized through Learn-Then-Test into a linguistically calibrated response.

Results show that reducing the specificity of the answers improves the factuality while still providing conformal guarantees.

**Questions:**

- There is a spelling mistake on Line 270: abstention instead of abstantion

**Ethical Concerns:**

["NO or VERY MINOR ethics concerns only"]

**Final Justification:**

The reason for my positive evaluation of the paper is due to its fresh perspective on uncertainty quantification through the lens of set prediction. This is supported by a well-motivated and suitable experimental setup.

I must note that I did have to re-read the sections where all the terminology and notations were introduced but in the end I thought it was well-written.

Though my question regarding different N samples was addressed, the rest of the points were left (justifiable) for future work. Given that I already had a positive score in combination with some points raised by other reviewers (e.g., the potential propagation of LLM errors throughout the pipeline and computational cost) I decided to keep my score the same.

**Limitations:**

A  discussion of the limitations and potential negative societal impact is missing / could be made more prominent.

**Quality:**

3

**Strengths And Weaknesses:**

**Strengths**
- Well written preliminary section to make the building stones clear enough to readers less familiar with possible world semantics and conformal prediction.
- Paper is also in general very well written and nicely structured.
- The selection of the datasets and experimental setup (e.g., metrics used, baselines compared to) is well-motivated. It is clear why SimpleQA (and Natural Questions) is chosen in the specific dataset, due to its  difficulty but high confident responses from models.
- The work targets an important problem in LLMs where models would still give a wrong response even when they are confident on an individual sample level.

**Weaknesses**
There are some analyses I thought were missing and would be really useful:
- An analysis on how the responses generated with Conformal Linguistic Calibration fare in terms of their verbalized confidence (e.g., if the answer is specific then the model is verbalized higher in confidence) would really strengthen the paper.
- How the performance varies across different amounts of N answers sampled from the LLM would be a very interesting insight.
- A user study would be very useful to showcase the utility of such a technique. Do users prefer being less specific to be more factually correct?

---

> ### Author Rebuttal · Authors · 2025-07-31
>
> Thank you for your encouraging review and recognition of our paper's clarity and structure. We appreciate your thoughtful questions about implementation details.
>
> ## Q1: CLC Generation and Verbalized Confidence
>
> Thank you for providing this interesting perspective. Through manual examination, we observe that our generated responses typically avoid explicit confidence statements. This is intended, as our formulation aims to represent uncertainty through hedging rather than verbalized confidence reports. In our implementation, our few-shot prompts in belief probing maintain structural similarity to original claims while incorporating appropriate hedging. We agree with the reviewer that it will be really interesting to see how LLM can simultaneously utilize different uncertainty communication protocols to improve user experience. **We will open-source all data and code to facilitate such investigations.**
>
> ## Q2: Performance Scaling with K
>
> Rerunning experiments with different K values requires expensive re-clustering and belief probing. However, we’d like to note that LTT remains valid with K=20. The effect of adding more number of samples is that it provides more fine-grained uncertainty level that the model can fall back to, as they produce more unique clusters:
>
> | K | Avg # Clusters |
> | --- | ------------- |
> |20 | 8.676 |
> |40 | 13.735 |
> |60 | 17.806 |
> |100 | 24.461|
>
> To summarize: Higher K provides more nuanced uncertainty control, but even K=20 maintains sufficient cluster diversity for effective calibration.
>
> ## Q3: Limitation Discussion and User Preferences
>
> You raise an important point about user perception. Our current limitation discussion focuses primarily on pipeline errors and guarantee validity. **We will expand this to include discussion of how AI confidence communication impacts human decision-making, acknowledging the challenges in conducting human-based factuality trade-off evaluations due to known biases (Li et al., 2025; Ding et al., 2025).**
>
> ## References
>
> - *Li, Jingshu, et al. "As Confidence Aligns: Understanding the Effect of AI Confidence on Human Self-confidence in Human-AI Decision Making." Proceedings of the 2025 CHI Conference on Human Factors in Computing Systems. 2025.*
> - *Ding, Yifan, et al. "Citations and trust in llm generated responses." Proceedings of the AAAI Conference on Artificial Intelligence. Vol. 39. No. 22. 2025.*

---

> > ### Comment · Reviewer_xhfo · 2025-08-05
> >
> > Thank you for addressing my concerns!
> >
> > Given my already positive viewpoint, I would like to maintain my score.

---

### Official Review · Reviewer_1Ph2 · 2025-07-03

**Clarity:** 2
**Significance:** 3
**Originality:** 3
**Rating:** 4
**Confidence:** 3

**Summary:**

This paper proposes Conformal Linguistic Calibration ( CLC ) framework to express uncertainty in language model outputs by controlling the level of linguistic generality in the outputs. This is achieved via generating increasingly less specific claims until a certain level of confidence is achieved. The authors reinterpret linguistic uncertainty through the lens of possible-world semantics and connect this idea with conformal prediction to provide probabilistic guarantees on factuality. Their method is instantiated via a Learn-Then-Test (LTT) framework and evaluated on two question answering datasets. The presented empirical results suggest that CLC surpasses traditional abstention-based methods in terms of informativeness and factual accuracy.

**Questions:**

Suggestions/Questions:

- Definition 2.1 and 2.2: These are difficult to parse due to the dense notation and minimal explanations. Could the authors clarify the key symbols and the relationships involved?

- Definition 3.1 and Theorem 3.2 : Hard to follow. Could you clarify these and provide intuitions and explanations ? What s the semantic meaning of the "sphere" in definition 3.1 and how should we interpret this in the context of language models? In theorem 3.2 what is the capital V represent? Is it equivalent to the sphere from Definition 3.2 ? if so this should be stated explicitly along with a explanation of what it means for a claim to describe such a sphere.

- Regarding utilizing an LLM for different components of the pipeline such as belief probing and nested cluster constructions, could you comment on how errors and hallucinations here would affect the pipeline and are there controls or mitigations strategies in the pipeline to correct these ?

On the experiments:
- K =100 samples per question introduces a significant computational overhead. In practice, many of these responses are duplicates and likelihood of observing novel responses diminishes rather quickly. How does the framework work under more constrained sampling budgets ? ( K =20 ? ) as commonly done in Uncertainty quantification literature?

The writing and presentation: The writing particularly sections 2.1 and 3 can be substantially improved. Many of the core theoretical contributions are stated without sufficient explanations. improving the clarify of these sections specially the definitions and formal statements would go a long way in improving clarity. I am open to revisiting my score if these are addressed.

**Ethical Concerns:**

["NO or VERY MINOR ethics concerns only"]

**Final Justification:**

After considering the rebuttal, I have increased my score from 3 to 4. The authors provided clarifications that helped resolve some of my earlier concerns, particularly regarding the related work.

However, some key concerns remain insufficiently addressed. In particular, I found the discussion around robustness to hallucinations across the multiple LLM components in the pipeline to be inadequate. My concern is that hallucinations originating in one module can propagate or compound in downstream components—potentially leading to information loss, the addition of irrelevant content during belief probing, or even the introduction of new information that was not part of the original LLM’s output distribution (and may affect evaluation as well). The authors did not adequately explore or acknowledge this issue in their response, which limits my confidence in the overall robustness and reliability of the proposed approach.

**Limitations:**

yes

**Quality:**

2

**Strengths And Weaknesses:**

Strengths:
- this paper addresses a very important issue in current LLM research which is how to convey the uncertainty in a manner that is both accurate and useful for end users. the problem of overconfident but incorrect answers is an important topic of research, and the paper takes a meaningful step towards addressing it.

- Connecting and combining possible-world-semantics from philosophy with conformal prediction techniques is indeed novel and interesting and can lead to meaningful avenue for future research as well.

- The presented experimental results seem promising.



Weaknesses

- The paper is difficult to follow ( specially section 2.1 and 3 ) due to unclear notation, undefined terms, and most importantly a lack of intuition behind many formal definitions ( e.g Definition 3.1, Theorem 3.2 ). Many claim/defs are introduced without sufficient explanations which makes it hard to follow.

- The empirical baselines are insufficiently justified. Many recent works also combine conformal prediction with abstention mechanisms for uncertainty calibration, and the paper should either compare to these methods or provide a clear argument for why a direct comparison is not meaningful. I believe a small related work section is also needed with such clarifications whic would improve clarity and highlight the novelty for the reader.


- the CLC pipeline has many moving parts ( belief probing, clustering, rewriting, nesting, calibration ) and a pseudocode or summary diagram mapping each component would be extremely helpful.

- It is unclear how robust the pipeline is to errors that is introduced by the LLM itself, particularly during belief probing and rewriting and how errors due to hallucinations in different parts of the pipeline could accumulate.

---

> ### Author Rebuttal · Authors · 2025-07-31
>
> Thank you for recognizing the importance of uncertainty communication in LLMs and our novel integration of possible-world semantics with conformal prediction. We address your concerns about clarity and technical presentation below.
>
> ## Q1: Clarified Explanation of Preliminaries
>
> **Definition of Sphere:** In possible world semantics, spheres represent different levels of certainty or likelihood around what we consider "normal" or expected. Possible worlds are organized into concentric spheres by accessibility relationships. A claim is **necessarily true** within a sphere if true in all accessible worlds, and **possibly true** if true in some accessible worlds. We’ll add more description in the revision with the additional page.
>
> **Intuitive Explanation:** We represent claims as sets of possible worlds where they remain true. By making claims less specific (hedging), we make them compatible with more possible alternatives, reducing factual error risk while maintaining informativeness. This formulation reveals connections between claim generation and set prediction, enabling conformal prediction for risk control. We will make sure that we add more substantial explanations and illustrations to explain this idea providing additional space.
>
> ## Q2: Related Work and Baseline Selection
>
> We have a related work section in **Appendix A**. Our approach differs fundamentally from standard abstention methods by enabling partial abstention at the claim level. While existing methods refuse to answer entirely, we can respond with partial knowledge when confidence is relatively low (see Figure 3 example).
>
> For baseline, we compare against the closest approach (Mohri and Hashimoto, 2024) in Section 4.1. While as mentioned by the reviewer there are other works that study LLM abstention and control the risk with conformal techniques, they mostly share a very similar structure to the baseline we compared to on short response QA (Yadkori et al., 2024). And from a previous study the sample-based uncertainty we adopt for our baseline is generally considered the best non-conformity score for this purpose (Cole et al., 2023). Our result clearly shows that standard abstention methods struggle under challenging settings like SimpleQA, where few prompts can be answered correctly with confidence. Also, at near-full abstention, their accuracy becomes unstable due to insufficient remaining examples.
>
> ## Q3: Procedure Diagram and Implementation Details
>
> **Figure 2 Clarification:** This serves as our system diagram, with subtitles corresponding to Section 3 paragraphs. We will align subtitle naming with paragraph headers and add clearer indications of the diagram's purpose.
>
> **Belief Probing Robustness:** In general LTT remains valid as long as we have an independent and identically distributed calibration set, even if our components are not always correct. On top of this, one of the authors conducted manual verification on over 50 examples from different prompt categories to see whether a unique answer is entailed by the summary. The manual verification study shows **precision: 0.81, recall: 0.91** across different question types, indicating robustness.
>
> ## Q4: Cluster Analysis with Varying K Values
>
> While it will be interesting to know how performance scales K, rerunning experiments with different K values requires expensive re-clustering and belief probing. However, we’d like to note that LTT remains valid with K=20. In general, the effect of adding more number of samples is that it provides more fine-grained uncertainty level that the model can fall back to, as they produce more unique clusters:
>
> | K | Avg # Clusters |
> | --- | ------------- |
> |20 | 8.676 |
> |40 | 13.735 |
> |60 | 17.806 |
> |100 | 24.461|
>
> To summarize: Higher K provides more nuanced uncertainty control, but even K=20 maintains sufficient cluster diversity for effective calibration.
>
> ## References
>
> - *Mohri, Christopher, and Tatsunori Hashimoto. "Language models with conformal factuality guarantees." Proceedings of the 41st International Conference on Machine Learning. 2024.*
> - *Yadkori, Yasin Abbasi, et al. "Mitigating llm hallucinations via conformal abstention." arXiv preprint arXiv:2405.01563 (2024).*
> - *Cole, Jeremy, et al. "Selectively Answering Ambiguous Questions." Proceedings of the 2023 Conference on Empirical Methods in Natural Language Processing. 2023.*

---

> > ### Comment · Reviewer_1Ph2 · 2025-08-05
> >
> > Thank you for your response and clarifications. Based on the rebuttal, I have raised my score accordingly.

---

### Official Review · Reviewer_YT5P · 2025-07-03

**Clarity:** 3
**Significance:** 4
**Originality:** 4
**Rating:** 5
**Confidence:** 4

**Summary:**

This paper proposes a novel paradigm for Large Language Models (LLMs) to communicate uncertainty called Conformal Linguistic Calibration (CLC). Instead of abstaining or using verbal hedges like "possibly," CLC enables a model to controllably reduce the specificity of its answer until it can guarantee its factuality with a certain probability. The method is grounded in possible world semantics and uses Conformal Prediction's Learn-Then-Test (LTT) framework to provide formal guarantees. The CLC pipeline involves sampling multiple answers, clustering them semantically, building nested sets of these clusters, and generating progressively vaguer claims that describe these sets. The authors demonstrate that CLC effectively trades informativeness for factuality, outperforming standard abstention baselines on challenging QA datasets. Finally, they show that this complex capability can be successfully distilled into a single, fine-tuned model that performs adaptive, uncertainty-aware claim rewriting.

**Questions:**

The "Belief Probing" step, where an LLM summarizes answer clusters into a vaguer claim, is very creative. How sensitive is the final performance to the specific prompt used for this step? Could a slightly different prompt lead to summaries that inadvertently break the nested property required for calibration, and does the LTT framework fully absorb this risk?

The computational cost of the pipeline is a notable limitation. The paper shows that the skill can be distilled into an 8B model. How well does this distilled rewriter generalize? For example, having been trained on QA data, can it effectively rewrite claims from out-of-distribution domains like scientific or legal text while maintaining the desired factuality-specificity trade-off?

Your method for constructing nested cluster sets prioritizes semantic similarity over the standard conformal approach of using model confidence scores (e.g., class probabilities). You justify this as helping generate more coherent summaries. Could you elaborate on the theoretical implications of this choice? Does this deviation from standard practice affect the tightness or validity of the LTT conformal guarantees in any way?

**Ethical Concerns:**

["NO or VERY MINOR ethics concerns only"]

**Final Justification:**

I keep my overall score at 5 (Accept). The rebuttal addressed my main questions about (i) robustness of the belief-probing step and the impact of occasional nesting violations—LTT does not require perfect nesting and the authors’ small manual check (P≈0.81/R≈0.91) suggests reasonable fidelity; (ii) practicality—showing that the expensive pipeline can be amortized via a distilled 8B rewriter; and (iii) the choice of semantic clustering over confidence scores—consistent with LTT since any nested family suffices for validity. I still see compute cost and multi-stage LLM dependence as limitations, but they are acknowledged and do not outweigh the contribution. No further issues remain.

**Limitations:**

Yes

**Paper Formatting Concerns:**

None.

**Quality:**

4

**Strengths And Weaknesses:**

1. Originality and Significance: The core idea is highly innovative and significant. It reframes the problem of LLM uncertainty in a practical and powerful way, moving beyond existing paradigms. The ability to explicitly control the trade-off between factuality and specificity, backed by formal guarantees, is a major step toward building more reliable and trustworthy AI systems.

2. Technical Quality: The work is technically strong, impressively unifying concepts from linguistic pragmatics, possible world semantics, and conformal prediction into a coherent framework. The experimental validation is comprehensive: it clearly demonstrates the factuality-specificity trade-off, shows superior performance over a strong baseline, and validates the approach by successfully distilling the complex pipeline into an efficient, instruction-following model.

3. Clarity and Impact: The paper is well-written, and the authors do an excellent job of explaining a complex, multi-stage process. The figures (especially the pipeline overview in Fig. 2) and results are clear and compelling. The release of the fine-tuned rewriting model and dataset is a valuable contribution to the community.

Weaknesses:

1. Pipeline Complexity and Cost: The main limitation is the computational expense and complexity of the CLC data generation pipeline. Requiring 100 sampled answers per question and multiple processing steps with a large "processor" LLM (GPT-40) creates a significant barrier for replication and adoption. While the distillation of the final model mitigates this for inference, the initial cost is very high.

2. Fragility and LLM-Dependency: The pipeline's quality is highly dependent on the capabilities of the LLMs used for intermediate steps like answer clustering and belief summarization. The authors astutely note that the summarization step is imperfect and use the LTT framework to account for this, but this reliance on prompted LLMs remains a point of potential fragility in the overall process.

---

> ### Author Rebuttal · Authors · 2025-07-31
>
> Thank you for your encouraging review and for speaking high of our work's originality and significance. We address your three specific questions (as well as related weaknesses) below.
>
> ## Q1: Belief Probing and Nested Property
>
> **Effectiveness of the current probing setup:** We thank the reviewer for raising the concern about potential errors in the pipeline and their implications. To address this, one of the authors conducted manual verification of whether a unique answer is entailed by the summary. The manual verification study shows **precision: 0.81, recall: 0.91** across different question types, indicating robustness of our current belief probing component.
>
> **Nested Property Violation:** As the reviewer correctly noted, relying on the LTT framework fully absorbs this risk—it doesn't require perfect nesting properties. Even if belief probing occasionally breaks nesting (producing irregular trade-offs), LTT's theoretical guarantees remain valid. Also empirically we observe more smooth and effective trade-off in Figure 3, and monotonic relationship between CPMI and multiplicity threshold levels, validating the effectiveness of our implementation.
>
> ## Q2: Distilled Model Generalization
>
> While providing explicit guarantees for OOD rewriting is challenging, our experiments suggest that the learned trade-off rewriting behavior has some generalizability. While our model is trained on QA data, the evaluation is conducted on Core (FActScore-like biographical claims) that is different from the QA data we trained, yet maintains effective factuality-specificity trade-offs. The distillation learns general hedging principles rather than domain-specific patterns. We’ll make this clearer in the revision. In theory, there are more recent conformal prediction methods that can deal with distribution shift explicitly, by modeling the shift directly (Prinster et al., 2024). So we can potentially leverage those work in the future. We do hope that future work will find better and more efficient rewriting that achieves a better tradeoff with higher correspondence.
>
> ## Q3: Semantic Clustering vs. Confidence Scores
>
> Using semantic similarity instead of confidence scores doesn't affect LTT guarantee validity. In fact any method that produces nested prediction sets is valid for conformal prediction, regardless of how those sets are constructed. As discussed by the reviewer, the major benefit of our approach is that it leads to more coherent nested sets, making belief probing easier and producing more interpretable claims.
>
> ## References
>
> - *Prinster, Drew, et al. "Conformal Validity Guarantees Exist for Any Data Distribution (and How to Find Them)." International Conference on Machine Learning. PMLR, 2024.*

---

> > ### Comment · Reviewer_YT5P · 2025-08-05
> >
> > Thank you for the detailed rebuttal. I have re-read the paper and the responses. My main questions on belief probing robustness, the nested property under LTT, and generalization of the distilled rewriter were addressed to my satisfaction. The remaining concern on compute cost is acknowledged and scoped as future engineering work. I am satisfied and keep my overall score at 5. I have no further questions for the authors.

---

### Note · Authors · 2025-08-13

We appreciate the insightful feedback from all reviewers. We're pleased that our formulation was found novel and engaging, and believe all concerns have been thoroughly addressed through multiple rounds of rebuttal process, as listed below:

**Methodology Clarification**: Following Reviewer 1Ph2's feedback, we've clarified our approach and will provide more intuitive explanations before technical details in the final version.

**Component Verification**: We conducted additional human verification experiments on belief summarization, analyzed how unique answers scale with K, and expanded our discussion of clustering approaches. Per reviewer's request our new comprehensive ablations during author-reviewer discussion should further validate our design choices regarding model choices, LLM-based clustering, and outer structure incorporation.

**Robustness**: We demonstrated that since the weak LTT assumptions, our framework maintains valid factual guarantees even with imperfect components.

**Related Work**: Per reviewer suggestions, we'll relocate the related work section from appendix to main body when space permits.

**Baseline Comparison**: We clarified that our baseline is the closest work to ours and represents the full spectrum of abstention-based approaches using sampling-based uncertainty quantification that has been reportedly effective. Our experiments show CLC's advantage stems from our fundamentally different formulation enabling factually correct responses despite limited knowledge.

**Generalizability**: The distilled claim rewriter, trained on QA data but evaluated on FActScore (report generation), demonstrates cross-domain performance, indicating it captures general hedging principles rather than domain-specific patterns.

**Future Work**: We acknowledge Reviewer xhfo's suggestions regarding user preference testing and cost reduction as valuable future directions and will include these in our limitations section.

We're committed to incorporating all agreed-upon revisions in the camera-ready version and thank the reviewers for their constructive guidance.

---

### Decision · Program_Chairs · 2025-09-17

**Decision:**

Accept (poster)

**Comment:**

As reviewers have mentioned, the problem studied in this paper is well-motivated, the ideas are interesting, empirical studies are convincing, and the paper is well-written. Given these positive elements, I recommend accept.